# First Case of Candida Auris Sepsis in Southern Italy: Antifungal Susceptibility and Genomic Characterisation of a Difficult-to-Treat Emerging Yeast

**DOI:** 10.3390/microorganisms12101962

**Published:** 2024-09-27

**Authors:** Stefania Stolfa, Giuseppina Caggiano, Luigi Ronga, Lidia Dalfino, Francesca Centrone, Anna Sallustio, Davide Sacco, Adriana Mosca, Monica Stufano, Annalisa Saracino, Nicolo’ De Gennaro, Daniele Casulli, Nicola Netti, Savino Soldano, Maria Faggiano, Daniela Loconsole, Silvio Tafuri, Salvatore Grasso, Maria Chironna

**Affiliations:** 1Microbiology and Virology Unit, Department of Interdisciplinary Medicine, University of Bari “A. Moro”, 70124 Bari, Italy; stefania.stolfa@policlinico.ba.it (S.S.); luigi.ronga@policlinico.ba.it (L.R.); adriana.mosca@uniba.it (A.M.); 2Hygiene Section, Department of Interdisciplinary Medicine, University of Bari “A. Moro”, 70124 Bari, Italy; giuseppina.caggiano@uniba.it (G.C.); sacco.davide19@gmail.com (D.S.); daniela.loconsole@uniba.it (D.L.); silvio.tafuri@uniba.it (S.T.); 3Hygiene Unit, Bari Policlinico University Hospital, 70124 Bari, Italy; francesca.centrone@policlinico.ba.it (F.C.); annasallustio@libero.it (A.S.); daniele.casulli@hotmail.com (D.C.); nettinicola@icloud.com (N.N.); 4Intensive Care Unit II, Department of Precision Medicine, Ionic Area, University of Bari “A. Moro”, 70124 Bari, Italy; lidia.dalfino@yahoo.com (L.D.); monistufa@gmail.com (M.S.); salvatore.grasso@uniba.it (S.G.); 5Clinic of Infectious Diseases, Department of Biomedical Sciences and Human Oncology, University of Bari “A. Moro”, 70124 Bari, Italy; annalisa.saracino@uniba.it (A.S.); nico84degennaro@gmail.com (N.D.G.); 6Policlinico Hospital Sanitary Direction, Bari Policlinico University Hospital, 70124 Bari, Italy; savino.soldano@policlinico.ba.it; 7Pharmacy Unit, Bari Policlinico University Hospital, 70124 Bari, Italy; maria.faggiano@policlinico.ba.it

**Keywords:** *Candida auris*, whole-genome sequencing, infection control, phylogenesis

## Abstract

*Candida auris* is an emerging yeast considered a serious threat to global health. We report the first case of *C. auris* candidemia in Southern Italy, characterized using whole genome sequencing (WGS), and compared with a second strain isolated from a patient who presented as *C. auris*-colonized following screening. The *C. auris* strain was isolated from clinical samples, identified via MALDI-TOF, and subjected to WGS. Antifungal susceptibility testing was performed using commercial broth microdilution plates, and resistance protein sequences were evaluated with TBLASTN-2.15.0. Following the initial *C. auris* isolation from patient A, active surveillance and environmental investigations were implemented for all ICU patients. Of the 26 ICU surfaces sampled, 46.1% tested positive for *C. auris* via real-time PCR. Screening identified a second patient (patient B) as *C. auris*-colonized. The phylogenetic characterization of strains from patients A and B, based on the D1/D2 region of the 28s rDNA and the internal transcribed spacer (ITS) region, showed high similarity with strains from Lebanon. SNP analysis revealed high clonality, assigning both strains to clade I, indicating a significant similarity with Lebanese strains. This case confirms the alarming spread of *C. auris* infections and highlights the need for stringent infection control measures to manage outbreaks.

## 1. Introduction

*Candida auris* is a multidrug-resistant (MDR) yeast of public health concern. Hospital-acquired infections (HAI) across five continents have been reported [1]. *C. auris* causes invasive infections particularly in immunocompromised patients, with high mortality rates mainly attributed to laboratory misidentification and MDR profiles. Furthermore, hospital outbreaks have been reported, probably due to its ability to create biofilms, making it resistant to antiseptics and persistent on environmental surfaces [2]. In 2019–2021, a large outbreak was reported in northern Italy with at least 277 cases [3] and a case fatality rate of 40.4% [4].

Six distinct clades of *C auris* have recently been reported [5]. These clades were distinguished on the basis of the analyses of whole genome sequencing (WGS) data and single nucleotide polymorphism (SNP) variation. Furthermore, these clades diverged in terms of antifungal susceptibility profiles, epidemic potential, and clinical manifestations [5].

We report the first case of *C. auris* candidemia in a critically ill patient admitted to a Intensive Care Unit (ICU) of southern Italy. The patient had not travelled abroad or had any other recent hospitalizations.

In April 2024, a critically ill patient was admitted to the ICU of the University Hospital Policlinico of Bari (patient A). One week later, he developed a bloodstream infection by New Delhi metallo-beta-lactamase (NDM) *Klebsiella pneumoniae* carbapenemase (KPC) and, despite prompt active antimicrobial treatment, refractory septic shock ensued. Daptomycin and caspofungin were empirically added but, unfortunately, the patient died within 48 h. Two days later, *C. auris* was isolated from their blood cultures.

Immediately, active surveillance of patients and ICU environmental surfaces was started, and infection control measures were strongly implemented. One of the sixteen screened patients (patient B) turned out to be colonized by *C. auris* in endotracheal aspirate, skin, and rectal swabs. He was admitted 2 weeks before patient A from an Albanian ICU and remained in good clinical condition until his last clinical evaluation in June 2024.

## 2. Materials and Methods

Blood cultures from patient A were performed as part of standard procedure for patients with sepsis. *C. auris* strain was isolated from blood culture after incubation on Sabouraud Dextrose agar (Biomerieux Inc., Lyon, France) for 48 h at 37 °C. Skin surveillance swabs (particularly of the axilla and groin) were collected by vigorously rubbing the tip of the swab over the indicated site at least 3–5 times. Skin, rectal swabs, and tracheobronchial aspirate (TBA) were inoculated on Sabouraud dextrose agar and incubated at 37 °C for 72 h. The identification of *C. auris* was performed through MALDI-TOF. Isolates of *C. auris* from patient A and patient B were subjected to WGS (whole genome sequencing) using the MiniSeq platform (Illumina Inc., San Diego, CA, USA). DNA was extracted and purified with QIAamp DNA Mini kit, according to the recommendations of the manufacturer. Purified DNA was measured with a Qubit 4.0 Fluorometer (Invitrogen by ThermoFisher Scientific, Waltham, MA; USA) using the double-stranded DNA (dsDNA). Nextera XT paired-end sequencing with a paired-end run of 2 × 150 bp was performed for library preparation. FastQC V0.12.1 software was used to perform quality control checks on the raw sequence data [6,7].

Read data were aligned against the reference genome of *C. auris* B8441 (GenBank assembly accession GCA_002759435.2) using BWA-MEM [8]. The consensus was generated using Samtools (version 1.20) and consensus was generated from BAM file based on the contents of the alignment records. The consensus was written as FASTA file [9]. BLAST was used to retrieve sequences with >90% identity from NCBI database, which were then added to the background genomic dataset after manual curation. The alignment of the core genome, the detection of recombination events, and the detection of single nucleotide polymorphisms (SNPs) were performed using Parsnp v1.2 [10]. Phylogenetic analysis was performed using RaxML8 and the TVM evolutionary model [11]. The phylogenetic tree was constructed using Mega 11.0.13 software with the neighbor-joining algorithm and annotated in iTOL [12].

Antifungal susceptibility testing was performed using commercial broth microdilution plates (Sensititre YeastOne, ThermoScientific, Waltham, MA, USA) according to the manufacturer’s instructions. As there are currently no established *C. auris*-specific susceptibility breakpoints, tentative breakpoints proposed by the CDC [13] were selected to define the clinical category of susceptibility.

A total of 26 surfaces in the ICU were sampled. Specifically, high-touch patient-area surfaces (multi-parameter touch screen monitors, assisted ventilation touch screen monitors, bed rails, infusion pumps, bed keypad, drug preparation shelf) and common ward surfaces (computer keyboards, ultrasound probes, telephone) were evaluated. The surfaces were sampled using sterile swabs soaked in distilled water. All swabs were transported to the laboratory in refrigerated containers at a temperature of +4 °C and were analyzed immediately upon arrival.

## 3. Results

*C. auris* strains from patient A (BA02) and patient B (BA03) were phylogenetically characterized on the basis of the D1/D2 region of the 28s rDNA and the internal transcribed spacer (ITS) region (Figure 1 and Figure 2).

The analysis showed a high degree of similarity of BA02 and BA03 with strains isolated in Lebanon (Bioproject PRJNA736600). Single nucleotide polymorphism (SNP) analysis showed that BA02 and BA03 had a high degree of clonality showing only 125 SNPs and were assigned to clade I, confirming a high rate of similarity with the strains collected in Lebanon (Figure 3). The difference in the number of SNPs between the BA03 strain of the present study and the Lebanon strain GCA019039495 was 381.

The sequences of the original clinical specimens described here have been deposited in the Genebank sequence database (accession number: SAMN41379019, SAMN41379020).

The isolates were both resistant to fluconazole and amphotericin B and susceptible to caspofungin, anidulafungin and, micafungin.

The evaluation of resistance protein sequences was performed using TBLASTN-2.15.0 [14]. Compared with the reference genome, the isolates BA02 and BA03 harbored Y132F azole resistance-associated mutation in ERG11 and mutations M192I in ERG4, K74E in CIS2, K52N in SNQ2, and S70R in FCY1. Finally, A583S in TAC1b and K719N in STE6 were detected (Table 1). Both isolates also expressed flucytosine resistance harboring, a single mutation in the FCY1 gene [14].

The detection of *C. auris* from surfaces was performed by the commercial *Auris* ID detection real-time PCR kit according to the instructions for use (Olm Diagnostics, Braintree, UK, distributed by LionDx, Pordenone, Italy). Samples were considered positive when they showed a ct value ≤ 40. Of the 26 environmental samples, 12 (46.1%) were positive for *C. auris*, in particular, swabs from multi-parameter monitors, infusion pumps, bed rails, and telephones.

## 4. Discussion

*C. auris* infections represent a major challenge due to its unfavorable antifungal resistance profile, with more than 40% expressing combined resistance to two or more classes of antifungals, severely limiting therapeutic options [15]. This characteristic may complicate the management of invasive *C. auris* infections such as candidemia, potentially contributing to high mortality and transmissibility. *C. auris* candidemia usually follows colonization; therefore, the early identification of colonized patients is critical to prevent invasive infection through interventions on modifiable predictors.

To our knowledge, this is the first isolation of *C. auris* in the whole of southern Italy. Similar cases of *C. auris* importation into Italy have been described previously [16], but this result confirms its alarming spread, especially in hospital settings worldwide. The cases of *C. auris* importation are united by prolonged hospitalization in countries with a high rate of multi-resistant organisms and the need for invasive medical procedures (central venous catheter, urinary catheter recent surgical procedure, mechanical ventilation).

The high rate of similarity of the two analyzed strains, showing only 125 SNPS, clearly suggests an epidemiological link between patient A and patient B. On the other hand, it was not possible to establish an epidemiological link between the strains of the present study and the Lebanese strains, with which the strains of the present study showed a high degree of similarity, nor to identify the source of patient B colonization. In fact, the spread of *C. auris* in Albania is not known, nor are the sequences of the strains isolated in that country available. It also remains undefined through which routes strains of Lebanese origin may have caused the fatal case and the colonization of the patient admitted from the Albanian ICU. It is of note that Y132F in the Erg11 gene was detected in both isolates, reflecting the resistance to azoles. Y132F is commonly detected among South Asian isolates, particularly Indian, Pakistani, and Chinese, and it is considered a clade-I-specific marker of resistance against fluconazole as well as in other Candida species, such as *C. parapsilosis* strains, which have the same Y132F mutation in the ERG 11 gene [15,17,18].

The mutations in ERG4 may be linked to reduced susceptibility to amphotericin B in *C. auris*, although their impact on structural changes and gene expression has not been investigated [14].

K74E was reported in all susceptible and resistant isolates, suggesting this may represent a polymorphism not connected with resistance and may not be under selective pressure from antifungals. TAC1b is a transcription factor controlling CDR1 expression in Candida species. Mutations in TAC1b were associated with increased azole resistance [19,20]. Finally, STE6 is an a-pheromone ABC family transporter that showed antimycotic responses [21].

Antifungal susceptibility testing confirmed resistance to fluconazole and amphotericin B, as well as susceptibility to echinocandins, which are still the first choice of treatment. This demonstrates that the combination of molecular characterization and antimycotic susceptibility testing represents the best possible strategy to address the challenge of diagnosis and the treatment of both symptomatic infections and asymptomatic colonization.

## 5. Conclusions

In conclusion, our data suggest the hypothesis of a strain imported from Albania/Lebanon. It is notable that the strain has different characteristics from the Greek one collected by Rimoldi et al. since the common mutations detected are S70R on gene FCY1 and Y132F on gene ERG11 [16]. This suggests the likely different origins and dynamics of the strains. The high transmissibility, the ability of *C. auris* to cause outbreaks, and the high frequency of antifungal resistance among clinical isolates highlight the need for the early identification and implementation of global surveillance programs. Monitoring resistance, coordinating local and international antifungal surveillance protocols, and developing novel diagnostic tests and antifungal drugs may play a crucial role in improving the clinical outcome of ICU patients. After the first *C. auris* isolation, immediate and accurate environmental investigation and disinfection, in combination with timely clinical surveillance and updated HAI prevention bundles, controlled the spread of *C. auris* and quickly extinguished the epidemic outbreak.

## Figures and Tables

**Figure 1 microorganisms-12-01962-f001:**
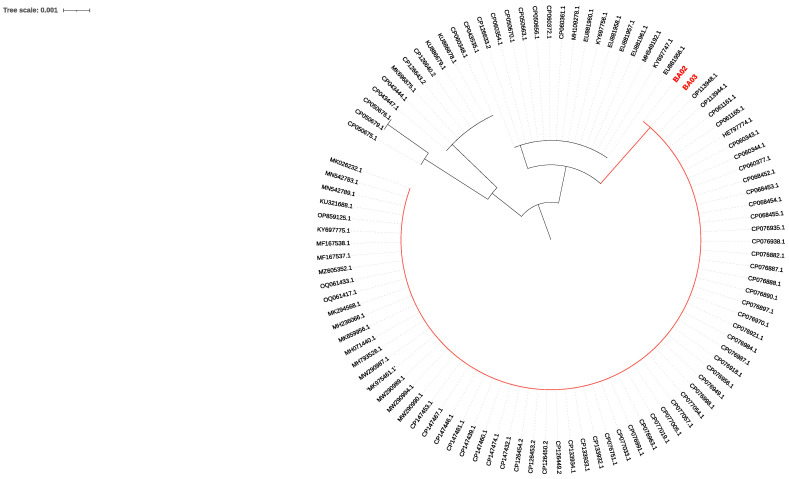
Neighbor-joining phylogenetic tree (max. seq. difference 0.75) on the partial D1/D2 large subunit (LSU) rDNA sequence of *C. auris* was constructed using 101 sequences. Sequences BA02 (from patient A) and BA03 (from patient B) are shown in red.

**Figure 2 microorganisms-12-01962-f002:**
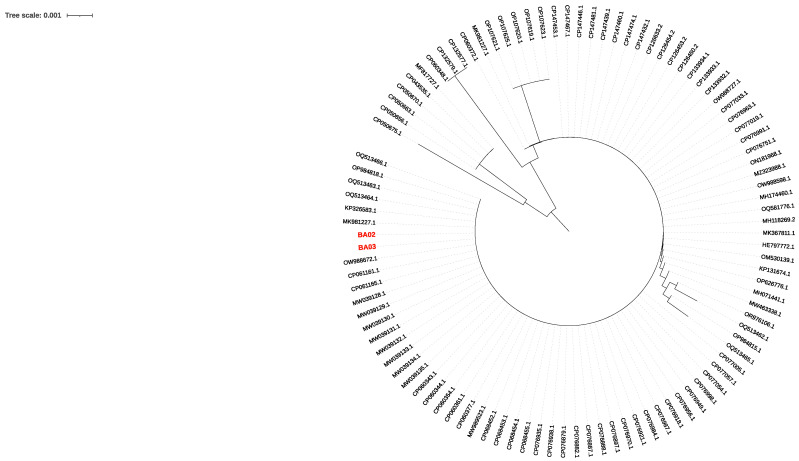
Neighbor-joining phylogenetic tree (max. seq. difference 0.75) on the internal transcribed spacer (ITS) sequence of *C. auris* was constructed using 101 sequences. Sequences BA02 (from patient A) and BA03 (from patient B) are shown in red.

**Figure 3 microorganisms-12-01962-f003:**
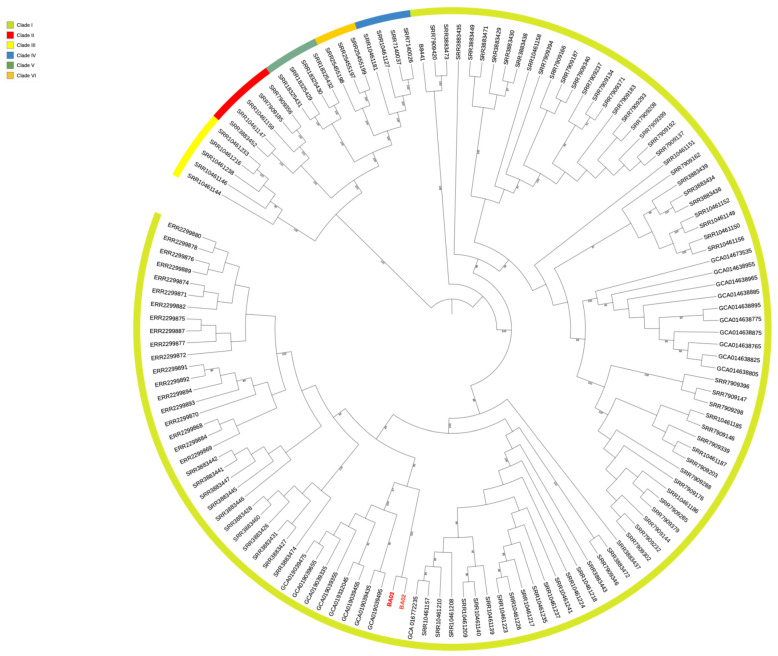
Six-clade population structure of *C. auris* using 139 genomes represented in the phylogenetic tree. The tree was generated by the maximum-likelihood method and annotated in iTOL. Numbers above branches are bootstrap values (only values >80 are shown). The strains of the present study are indicated in red.

**Table 1 microorganisms-12-01962-t001:** Antifungal susceptibility testing of the two *C. auris* isolates (upper part) and the mutations detected (lower part). * N/A: data not available.

MIC	Patient A		Patient B		CDC’s Tentative MIC Breakpoints
Amphotericin B	2	R	4	R	≥2
Anidulafungin	0.12	S	0.12	S	≥4
Caspofungin	0.12	S	0.12	S	≥2
Fluconazole	128	R	256	R	≥32
Isavuconazole	0.06	N/A *	0.12	N/A	N/A
Itraconazole	0.12	N/A	0.25	N/A	N/A
Micafungin	0.12	S	0.12	S	≥4
Posaconazole	0.06	N/A	0.06	N/A	N/A
Voriconazole	0.5	N/A	0.5	N/A	N/A
Mutations					
Gene		Gene ID		Substitution	
*CIS2*		B9J08_01093		K74E	
*ERG4*		B9J08_00711		M192I	
*SNQ2*		B9J08_03375		K52N	
*STE6*		B9J08_05399		K719N	
*ERG11*		B9J08_03698		Y132F+	
*TAC1B*		B9J08_04780		A583S	
*FCY1*		PIS48695.1		S70R+	

## Data Availability

Data will be available upon reasonable request by e-mail to Maria Chironna (maria.chironna@uniba.it).

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
