# Peer review of "First Case of Candida Auris Sepsis in Southern Italy: Antifungal Susceptibility and Genomic Characterisation of a Difficult-to-Treat Emerging Yeast"

_microorganisms, 2024, doi:10.3390/microorganisms12101962_

Round 1
Reviewer 1 Report
Comments and Suggestions for Authors
The Article was written in a clear and objective manner. It's very organized. But I have some important suggestions.
In lines 57 and 71 write the meaning of the acronyms that are highlighted.
In lines 105 and 108 specify BA02 and BA03.
Which sequence is from patient A or B? What about the other sequences in the figure? Specify.
On line 111 add a space at the beginning of the phrase "Single nucleotide..."
In line 116 Figure 3 add clade V and VI (In 2022 confirmation of the fifth Candida auris clade [doi: 10.1080/22221751.2022.2125349]; In 2024 sixth Candida auris clade in Singapore [doi: 10.1016/S2666-5247(24)00101-0].
Could improve the discussion.
Author Response
Comments to the Author
The Article was written in a clear and objective manner. It's very organized. But I have some important suggestions.
Response: We thank the reviewer for the encouraging comments.
Comment 1. In lines 57 and 71 write the meaning of the acronyms that are highlighted.
Response: Thank you. We have now resolved the acronyms issue (page 2, line 61, page 2, lines 65-66 and page 2, line 81).
Comment 2. In lines 105 and 108 specify BA02 and BA03
Response: We thank the Reviewer. The name of the strains from patients have been specified. (page 3, line 173).
Comment 3. Which sequence is from patient A or B? What about the other sequences in the figure? Specify
Response: Thank you. In the captions of Figures 1 and 2 the names of the strains and the corresponding patient has been specified (page 3, lines 178-179; page 4, lines 185-186).
Comment 4. On line 111 add a space at the beginning of the phrase "Single nucleotide..."
Response: Thank you. Done.
Comment 5. In line 116 Figure 3 add clade V and VI (In 2022 confirmation of the fifth Candida auris clade [doi: 10.1080/22221751.2022.2125349]; In 2024 sixth Candida auris clade in Singapore [doi: 10.1016/S2666-5247(24)00101-0].
Response: Thank you. The presence of six clades was mentioned in the Introduction section (page 2, lines 55-59). Figure 3 has been modified so that the six clades are now clearly labelled (page 5, line 196).
Comment 6. Could improve the discussion.
Response: We thank the reviewer for the advice. The Discussion section has been improved, as suggested (page 7, lines 243-249, lines 251-256, lines 257-265 and lines 280-285).
Reviewer 2 Report
Comments and Suggestions for Authors
Candida auris is important fungal pathogen, and the report of its infection in southern Italy is significant. However, this report was poorly written and cannot be published in its current form.
Major comments:
1. there was no report on the number of SNPs between the two samples, and other isolates from NCBI. Without the SNPs or genetic distance, there was no evidence to support for clonality. The authors indicated the origin to be from Lebanon. But without comparing to global isolates but simply focusing on Lebanon samples, it was hard to justify.
2. The method section was incomplete and lacked lots of information. Without detailed methods, there was no credibility in the data obtained for results/discussion.
Specific:
L37-38, What was the number of SNPs? How was analysis performed?
L73-75, How was DNA extracted? I am not aware of any 2x100 paired ends cycles available for Illumina Miniseq. Please double check.
L79-80, The description was inaccurate. What "consensus" was generated from samtools? which command/ criteria was used in samtools?
L80-81, I presume genome in fasta file format was searched against NCBI database using blast. But how was the fasta file generated. There was no description. Which genomes were retrieved from NCBI? What was the criteria in selecting these genomes. There was > 3000 C. auris genomes in NCBI.
L85, Phylogenetic analysis was performed on which dataset?
L101-109, How did you determine the LSU and ITS sequences? Fragment size? and how were NCBI sequences selected for analysis?
L110-112, What analysis had been performed? and what are variation or similarity? High clonality based on what? and how did you suggest the isolates from Lebanon?
L128, having a mutation did not necessarily mean developing resistance.
L136, how was real time PCR performed? Which assay/loci? What Ct values for cutoff (in positive samples)? There was no data at all.
L145-146, whole genome sequencing data only cannot establish the epidemiological links. Other information such as date of isolation, length of stay, ward, procedures .. etc are required.
Comments on the Quality of English Language
English is okay.
Author Response
Comments to the Author
Candida auris is important fungal pathogen, and the report of its infection in southern Italy is significant. However, this report was poorly written and cannot be published in its current form.
Response: We thank the reviewer for the comments. We have tried to improve the manuscript taking into account your valuable comments and suggestions.
Comment 1. there was no report on the number of SNPs between the two samples, and other isolates from NCBI. Without the SNPs or genetic distance, there was no evidence to support for clonality. The authors indicated the origin to be from Lebanon. But without comparing to global isolates but simply focusing on Lebanon samples, it was hard to justify.
Response: Thank you. The SNPs difference between samples BA02 and BA03 in the present study was 125 nucleotides and supports clonality. This was specified in the text (page 7, lines 261-262). We did not state that the origin was Lebanese, although the analysis showed that the closest strains were Lebanese, as 381 SNPs were found between BA03 and the Lebanese strain GCA019039495. This was specified in the text (page 4, lines 189-192 in the Results and page 7 lines 257-265 in the Discussion section).
Comment 2. The method section was incomplete and lacked lots of information. Without detailed methods, there was no credibility in the data obtained for results/discussion.
Response: We thank the Reviewer. The Materials and Methods section has been revised to indicate more clearly the analysis methods and tools used for sequence analysis.
Comment 3. Specific:
L37-38, What was the number of SNPs? How was analysis performed?
L73-75, How was DNA extracted? I am not aware of any 2x100 paired ends cycles available for Illumina Miniseq. Please double check.
L79-80, The description was inaccurate. What "consensus" was generated from samtools? which command/ criteria was used in samtools?
Response: We thank the Reviewer. The Materials and Methods contains the method used for DNA extraction (page 2, lines 84-88). Nextera XT paired-end sequencing with a paired-end run of 2 × 150 bp was performed for library preparation. The text was modified (page 2, line 89).
Consensus was generated from BAM file based on the contents of the alignment records. The consensus was written as FASTA file. Then command was: Samtools consensus -f fasta. The text was modified (page 2, lines 93-95).
L80-81, I presume genome in fasta file format was searched against NCBI database using blast. But how was the fasta file generated. There was no description. Which genomes were retrieved from NCBI? What was the criteria in selecting these genomes. There was > 3000 C. auris genomes in NCBI.
Response: We thank the Reviewer. Yes, the genomes in the fasta format were searched in the NCBI database. For the ITS and LSU analysis, 101 sequences were retrieved from Blast. The criterion used was to retrieve sequences with >90% identity. For the construction of the phylogenetic tree in Figure 3, 139 sequences defining the 6 currently identified clades were used. The text was modified accordingly (page 3, lines 178-179 figure 1; page 4, lines 185-186 figure 2 ; page 5, line 196 figure 3).
L85, Phylogenetic analysis was performed on which dataset?
Response: We thank the Reviewer. Phylogenetic analysis was performed retrieving sequences from NCBI. For the ITS and LSU analysis, 101 sequences were retrieved from Blast. The criterion used was to include sequences with >90% identity. For the construction of the phylogenetic tree in Figure 3, 139 sequences defining the 6 currently identified clades, were used (page 5, line 196). This was stated clearly in the text, as previously written.
L101-109, How did you determine the LSU and ITS sequences? Fragment size? and how were NCBI sequences selected for analysis?
Response: We thank the Reviewer. The LSU and ITS sequences were determined on the basis of the article by Rimoldi et al. published in 2024 (see References section). The length of the LSU was 560 nucleotides, while the length of the ITS was 425 nucleotides.
L110-112, What analysis had been performed? and what are variation or similarity? High clonality based on what? and how did you suggest the isolates from Lebanon?
Response: We thank the Reviewer. Genomes in fasta format were searched in the NCBI database. For ITS and LSU analysis, 101 sequences retrieved from Blast were used. The criterion used was to retrieve sequences with >90% identity. For the construction of the phylogenetic tree in Figure 3, 139 sequences defining the 6 currently identified clades were used. We did not state that the origin was Lebanese, but the analysis showed that the closest strains were Lebanese, as 381 SNPs were found between BA03 and the Lebanese strain GCA019039495. This was specified in the text (page 4, lines 189-192, and page 7, lines 257-265).
L128, having a mutation did not necessarily mean developing resistance.
Response: We thank the Reviewer. We agree. In the text we simply reported the common mutations that confer genomic resistance, as reported by Abid FB et al. 2023 (Molecular characterisation of Candida auris outbreak isolates in Qatar from patients with COVID-19 reveals the emergence of isolates resistant to three classes of antifungal drugs. Clin Microbiol Infect 2023). The specific reference has been added (page 9, lines 441-444).
L136, how was real time PCR performed? Which assay/loci? What Ct values for cutoff (in positive samples)? There was no data at all.
Response: We thank the Reviewer. Isolates of C. auris from patient BA02 and BA03 were available for whole genome sequencing. Environmental samples were tested by commercial real time PCR targeting the 28s rDNA region. Samples were considered positive for C. auris when they showed a ct value ≤40. This is now specified in the text (page 6, lines 229-232).
L145-146, whole genome sequencing data only cannot establish the epidemiological links. Other information such as date of isolation, length of stay, ward, procedures .. etc are required.
Response: We thank the Reviewer. We agree. The linkage was established taking into account the two final paragraphs of the Introduction section (page 2, lines 63-74). “Sample information (date of sampling, ward, type of specimen, testing results) together with the data of patients for whom molecular testing was performed (i.e. age and sex) were recorded in an anonymous database by changing sensitive data into alphanumeric code. No clinical data associated with these specimens were available” was stated in the Patient section (page 8, lines 385-392). Nevertheless, date of sampling, date of isolation etc. were available in Bioproject (accession number of sequences: SAMN41379019, SAMN41379020). In addition, in the section Data availability Statement is written that “Data will be available on reasonable request, by e-mail to Prof. Maria Chironna (maria.chironna@uniba.it)” (page 8, lines 393-394).
Round 2
Reviewer 1 Report
Comments and Suggestions for Authors
The necessary modifications have been made. Manuscript approved.